# Siglecs as Therapeutic Targets in Cancer

**DOI:** 10.3390/biology10111178

**Published:** 2021-11-13

**Authors:** Jackwee Lim, Duygu Sari-Ak, Tanaya Bagga

**Affiliations:** 1Singapore Immunology Network, A*STAR, 8a Biomedical Grove, Singapore 138648, Singapore; tanaya_bagga@immunol.a-star.edu.sg; 2Department of Medical Biology, School of Medicine, University of Health Sciences, Istanbul 34668, Turkey; duygu.sariak@sbu.edu.tr

**Keywords:** Siglec, sialic acid, cancer, immunosuppressive, anti-Siglec, treatment

## Abstract

**Simple Summary:**

Hyperglycosylated cancer cells are often decorated with abundant sialic acids, which are recognized by Sialic acid binding immunoglobulin type lectins (Siglecs) expressed on immune cells. As sialic acids are normally expressed on almost all cell types, Siglecs can function as receptors for “self”. Presently, 15 human Siglecs (including non-sialic acid binding Siglec XII) are known, of which most are immunoinhibitory receptors that induce strong inhibitory signaling when Siglecs bind sialic acids. Hence, tumor cells use hyper sialic acid expression in their favor to modify the immune system that can lead to immune suppression. Such engagement along the Siglec-sialic acid axis can dampen major killing modes of effector immune cells and suppress the activation of immune responses, which can lead to immune tolerance and tumor growth. In this review, we emphasized recent studies on all 15 Siglecs found in humans, many of which still remain understudied. In addition, we highlighted different strategies in disrupting the surface Siglec-sialic acid interactions, including antibodies and glycan moieties, which can steer back antitumor immune responses to reduce tumor size and improve overall survival of cancer patients.

**Abstract:**

Hypersialylation is a common post-translational modification of protein and lipids found on cancer cell surfaces, which participate in cell-cell interactions and in the regulation of immune responses. Sialic acids are a family of nine-carbon α-keto acids found at the outermost ends of glycans attached to cell surfaces. Given their locations on cell surfaces, tumor cells aberrantly overexpress sialic acids, which are recognized by Siglec receptors found on immune cells to mediate broad immunomodulatory signaling. Enhanced sialylation exposed on cancer cell surfaces is exemplified as “self-associated molecular pattern” (SAMP), which tricks Siglec receptors found on leukocytes to greatly down-regulate immune responsiveness, leading to tumor growth. In this review, we focused on all 15 human Siglecs (including Siglec XII), many of which still remain understudied. We also highlighted strategies that disrupt the course of Siglec-sialic acid interactions, such as antibody-based therapies and sialic acid mimetics leading to tumor cell depletion. Herein, we introduced the central roles of Siglecs in mediating pro-tumor immunity and discussed strategies that target these receptors, which could benefit improved cancer immunotherapy.

## 1. Introduction

Siglecs (Sialic acid-binding Immunoglobulin-like Lectins) might hold the key to better solutions for human diseases, including cancer. Recent studies on Siglecs have shown their important roles in tumor immunosurveillance including immunosuppression, making them attractive anti-cancer molecular targets. Presently, tumor immune escape is a hallmark of tumor growth [1] and involves tumor expression of inhibitory immune checkpoints or “don’t eat me” signals such as programmed cell death ligand 1 (PD-L1), β2-microglobulin (β2 m), CD24, and CD47 to evade immune responses [2,3,4,5]. Additionally, the loss of β2 m represents resistance to checkpoint blockades cytotoxic T lymphocyte-associated protein 4 (CTLA-4) and (programmed cell death protein 1) PD-1 blockade in cancers [6]. Moreover, anti-PD-1/PD-L1 treatment is not feasible for all patients [7,8], suggesting a need to identify other targetable immune checkpoints. 

Siglecs as potential immune checkpoints are expressed on most white blood cells in the immune system (Figure 1) and bind to sialoglycans with variable binding preferences [9]. However, sialic acid is not always needed for Siglec-ligand interaction, suggesting direct protein ligands [4]. Siglecs are categorized into CD33-related Siglecs that exhibit high sequence identity (50–99%) and the others (Siglec-1, Siglec-2, Siglec-4, and Siglec-15), sharing 25–30% sequence identity. Typically, all Siglecs consists of two distinct domains: the extracellular immunoglobulin (Ig) domains including variable (V) and constant (C2)-set domains, and a transmembrane domain. Additionally, among the 15 human Siglecs, most Siglecs bear one or more cytosolic immune receptor tyrosine-based inhibitory motifs (ITIMs) to signal negatively to dampen immune response. Whereas, certain Siglecs (Siglec-14, -15, -16) do not have intracellular domains but bear certain basic amino acids, which recognize tyrosine-based activation motif (ITAM)-encoding molecules (e.g., DAP-12) for signal activation [10]. The roles of Siglecs in various diseases have been highlighted by many excellent reviews (Figure 1) [11,12,13,14,15,16,17,18,19,20,21]. As current cancer immunotherapy focuses on selected and upregulated targets instead of systemic activation of immune responses, herein we summarized the latest findings and strategies on human Siglecs as targets in cancer, which are immune normalization targets in glyco-immunology. 

## 2. The Siglec Family

### 2.1. Siglec-1 (CD169)

Siglec-1, also known as Sialoadhesin, is found only in macrophages, monocyte precursors, and mature dendritic cells. Due to its lack of any tyrosine-based signaling motifs, its predominant role is cell adhesion via high affinity ligand NeuAcα2-3Gal on N- and O-linked glycans and glycolipids [23]. Siglec-1 consists of one ligand-binding V-set domain and 16 C2-set domains, which extend its binding site away from the cell surface. Besides classical M1 or M2 type macrophage classification, a CD169^+^ macrophage subpopulation is found in the bone marrow, lymph node, liver, and spleen [24,25]. Despite the lack of a signaling motif, CD169^+^ macrophages still play multiple roles in cancer progression [26]. CD169^+^ macrophages represent a heterogeneous population including tumor infiltrating macrophages (TIMs) and tumor-associated tumor macrophages (TAMs) that can have anti- and pro-tumor functions, respectively. Besides TIMs and TAMs, sinus CD169^+^ macrophages in the tumor-draining lymph nodes are widely regarded in establishing antitumor immunity. Sinus CD169^+^ macrophages interact via cross-antigen presentation with basic leucine zipper ATF-like transcription factor 3 (BATF3)-dependent dendritic cells for the activation of CD8^+^ T cells [27,28]. Similarly, many studies have shown that abundant CD169^+^ macrophages in lymph nodes are positively correlated to the amount of CD8^+^ T cells in the tumor, and favorable patient overall survival (OS) in bladder cancer, colon cancer, gastric cancer, and malignant melanoma [26,29,30,31] However, Singh and Choi recently unraveled that sinus CD169^+^ macrophages can promote the survival of melanoma cells via mitotic polo-like kinase 1 (PLK1) phosphorylation at the subcapsular sinus region [32]. In comparison, CD169^+^ TIMs found in human hepatocellular carcinoma tissue are positively correlated with better patient’s OS but not non-tumoral CD169^+^ cells [33]. Conversely, CD169^+^ TAMs are immunosuppressive, and were recently shown to display elevated PD-L1 expression due to activated JAK2 driven by granulocyte colony-stimulating factor (G-CSF) level in triple-negative breast cancer [34]. Therefore, the inhibitory PD-L1/PD-1 axis may be exploited by breast cancer cells to escape immune surveillance, whereas CD169^+^ TAM depletion has shown infiltration of cytotoxic CD8^+^ T cells in the breast and lung tumor microenvironment [34]. 

### 2.2. Siglec-2 (CD22)

Siglec-2 belongs to a family of immune inhibitory Siglecs (Siglecs-2, -3, -4, -5, -6, -7, -8, -9, -10, -11, and -12) which bears ITIM to deliver immunosuppressive signals such as reduced phagocytosis, dampened inflammation, and inhibited danger-associated molecular pattern/ pathogen-associated molecular pattern (DAMP/PAMP)-mediated inflammation. Unlike Siglec-1 found in myeloid cells that preferentially binds to α 2,3-linked sialylated glycans, Siglec-2 is exclusively expressed on mature B cells, and strongly binds to α 2,6-linked sialylated ligands such as N-acetylneuraminic acid, 2,6 galactosides (Neu5Ac, α(2-6) Gal), and N-glycolylneuraminic acid, 2,6 galactosides (Neu5Gc, α(2-6) Gal) [35]. The structure of Siglec-2 contains six tyrosines in its cytoplasmic tail, four of which are in ITIM motifs: Y783, Y843, Y863, and Y828. Following cross-linking of B-cell receptors (BCRs), these tyrosine residues are phosphorylated and recruit Src homology region 2 (SH2) domain-containing protein tyrosine phosphatase-1 (SHP-1), which dephosphorylates BCR-proximal signaling complexes [36]. Additionally, Src homology region 2 (SH2) domain-containing inositol 5’-phosphatase (SHIP) can form a quaternary SHIP-Grb2-Shc-CD22 complex, which negatively regulates calcium influx (Figure 2) [37]. Like Siglec-G, Siglec-2 interacts with BCR and inhibits BCR signaling [38]. Moreover, both Siglec-G and Siglec-2 are also associated with toll-like receptor signaling, bridging adaptive and innate immune responses [35]. Notably, the restricted expression of Siglec-2 on B cells and its inhibitory role in dampening BCR activation make it an ideal target in B-cell malignancies (Figure 2). Unlike CD19-targeting chimeric antigen receptors (CAR) T cell therapy with acquired resistance due to surface CD19 antigen loss [39,40], CD22 usually retains expression in CD19_neg_ tumors [41,42]. As such, CD22 or tandem CD19 and CD22 targeting CAR-T cells are being pursued in treating B cell malignancies such as non-Hodgkin lymphomas and acute lymphoblastic leukemia [43,44,45]. However, even CD22 expression can be low or even null in several human lung cancer cells, including pancreatic, prostate, breast, and liver cancers to be considered targetable (https://www.proteinatlas.org/ENSG00000012124-CD22/pathology, accessed on 3 November 2021) [46]. More importantly, CD19/CD22 CAR-T cell therapy can lead to hemophagocytic lymphohistiocytosis-like toxicities due to persistent elevation of inflammatory cytokines in concert with pronounced NK-cell lymphopenia [47,48]. 

### 2.3. Siglec-3 (CD33)

Siglec-3 is expressed on the surface of myeloid progenitor cells, monocytes, macrophages, and microglia. Unlike Siglec-1, the smaller Siglec-3 does not protrude out from the cell surface. It consists of a V-set Ig-like domain, a C2-set Ig-like domain, a transmembrane domain, and a cytoplasmic tail bearing two conserved ITIMs, which is shared across all CD33-related Siglecs. However, CD33 is the only Siglec with a peroxisome-targeting sequence in the cytoplasmic tail [49,50]. Alternate splicing of the Siglec-3 RNA can form a shorter isoform (CD33 m) that lacks the V-set Ig-like domain [49], resulting in its inability to bind to α2-3-, α2-6-sialylated glycans and leukemia cell lines [51]. Hence, CD33 m cannot localize to the cell surface and is retained in the peroxisomes in blood neutrophils, monocytes, and microglia [50]. Like other myeloid inhibitory Siglecs (such as Siglec-7 and -9), Siglec-3 has a downstream signaling function similar to other ITIM-containing immune checkpoint receptors like PD-1 and CTLA-4 [52,53], which are popular checkpoint targets for cancer treatment [49]. Therefore, CD33 is a target for antagonistic mAbs and CAR-T cell immunotherapy, particularly in adult and pediatric acute myeloid leukemia (AML) [51,54]. However, CD33 is a challenging target due to its low cell surface density, slow antibody-induced internalization, and expression on hematopoietic progenitor cells [55,56], whereby non-specific targeting can potentially lead to cytopenia [49]. Therefore, targeted immunotherapy based on antibody-drug conjugates targeting AML has been limited to CD33-targeting gemtuzumab ozogamicin (Mylotarg^TM^) [56,57]. More recently, to overcome resistant AML tumors, ^225^Ac-conjugated anti-CD33 (Actinium-225-lintuzumab) has been used in tandem with B cell lymphoma 2 selective inhibitor venetoclax to reduce antiapoptotic myeloid cell leukemia 1 overexpression [58]. 

### 2.4. Siglec-4 (Myelin Associated Glycoprotein, MAG)

Native Siglec-4 or MAG has five Ig-like domains: one N-terminal V-set domain, and four C2-set domains only expressed by oligodendrocytes in the central nervous system (CNS), and Schwann cells in the peripheral nervous system [56]. Siglec-4 exists in two splice variants, S-MAG and L-MAG, and contains a tyrosine phosphorylation site [59,60]. Among all Siglecs, Siglec-4 is the most conserved across all vertebrates and preferentially binds to sialic acid α(2-3)-linked to D-galactose [60]. Similar to Siglec-1 found in microglia but not Siglec-2 [11], Siglec-4 has low affinity towards N-glycolylneuraminic acid (Neu5Gc) [60]. Siglec-4 is prominently known as a cell adhesion molecule carrying the human natural killer 1 (HNK-1) carbohydrate, a sulfated trisaccharide (HSO_3_-3GlcAβ1-3Galβ1-4GlcNAc) in the nervous system. Additionally, autoantibodies against myelin and MAG have been reported in several neuropathic syndromes, including anti-MAG neuropathy with distinct HNK-1 epitopes [56,61,62]. More interestingly, besides the brain, the interaction between MAG and cancer-associated MUC-1 is found to be sialic-acid dependent in pancreatic cancer perineural invasion [63]. Additionally, in CAR functionality, a Siglec-4-derived long spacer has similar antitumor efficacy as a CD8α spacer for CAR-T therapy but is proposed to better target membrane-proximal epitopes with minimized CNS toxicities [64].

### 2.5. Siglec-5 (CD170) and Siglec-14 Pairs

Siglec-5 and Siglec-14 are Siglec pairs. Siglec-5 is expressed on neutrophils, monocytes, B cells, mast cells, and activated T cells, and recognizes a wide range of sialic acid structures, including α2-3, α2-6, and α2-8-linked sialyllactosamines [65]. As Siglec pairs, both Siglec-5 and -14 are highly homologous, typically expressed together, and recognize similar sialylated ligands [66]. However, Siglec-5 bears an inhibitory ITIM motif to counterbalance activating Siglec-14, which recruits DNAX activation protein of 12 kDa (DAP12) bearing the ITAM motif [66]. Notably, tyrosine phosphorylation of Siglec-5 is not needed for the recruitment of SHP-1 and SHP-2 to mediate inhibitory activity [67]. Siglec-5 and its soluble form (sSiglec-5) are also known to associate with leukocyte counter-receptor P-selectin glycoprotein ligand-1, prevent leukocyte recruitment to sites of inflammation [68], and maintain a pro-tumoral environment. Previously, in some acute myeloid leukemia cases, aberrant co-expression of Siglec-5 and CD34 was observed in blasts [69]. More recently, sSiglec-5 was shown to be prognostic for colorectal cancer [70]. More interestingly, resting T cells normally express few or no Siglecs, but Siglec-5 is found expressed on activated tumor infiltrating T lymphocytes of various cancers, which can lead to T cell exhaustion [13,71,72]. As Siglec-5 in human cells is strongly repressed, it still remains unknown what exactly triggers Siglec-5 expression in T cells in cancers [13]. On the other hand, Siglec-14 is highly homologous to inhibitory Siglec-5 and found on granulocytes and monocytes but not B cells [73]. As an activating Siglec, it enhances inflammasome activation and macrophage IL-1β release, which is counteracted by Siglec-5 [74]. Like sSiglec-5, soluble Siglec-14 derived from mRNA splicing is detected in the blood and suppressed pro-inflammatory responses of membrane-bound Siglec-14^+^ THP-1 leukemia cell lines [75]. However, Siglec-14 currently remains understudied in cancer.

### 2.6. Siglec-6 (CD327)

Siglec-6 is predominantly expressed by mast cells, B cells, and a minor AS-type dendritic cell subset [76,77,78]. Functionally, Siglec-6 is an immune-inhibitory CD33-related Siglec and strongly binds to sialylated Tn-structures [79]. As an exception, all Siglecs recognize the carboxyl group of sialic acid, but only Siglec-6 does not require the glycerol side-chain for binding [65]. In cancer, Siglec-6 was recently found to be upregulated in circulating and urinary CD8^+^ T-cell of non–muscle-invasive bladder cancer patients, wherein a high level of Siglec-6 was associated with advanced bladder cancer [80]. Additionally, Siglec-6 is prevalently expressed on acute myeloid leukemia blasts and transformed B cells in chronic lymphocytic leukemia [78]. As Siglec-6 mRNA and protein are not expressed in hematopoietic stem cells, they are novel targets for CAR-T cell immunotherapy in chronic lymphocytic leukemia [81].

### 2.7. Siglec-7 (p75/AIRM1, CD328)

Siglec-7 is a natural killer (NK) cell-inhibitory receptor bearing ITIM motifs and is mainly expressed on NK cells, monocytes, macrophages, mast cells, neutrophils, dendritic cells, and a minor subset of CD8^+^ T cells but not B cells [82,83,84]. Among all Siglecs, Siglec-7 has very high affinity towards α2,8-linked disialyl residues and is found to have two sialic acid binding sites instead of one [85,86]. The natural Neu5Acα2,8-Neu5Ac glycans are lowly expressed on T cells and monocytes but highly expressed on NK cells, which can be masked via *cis* interactions [84]. More recently, synthetic Neu5Ac-edited NK was found to remodel glycans via transfer to tumor cells, which accumulated the sialylated glycans [87]. Accumulating studies have also reported that tumor cells overexpress Siglec-7 ligands to evade NK cell lysis [88]. Upon cognate ligand binding, the recruitment of SHP-1 and SHP-2 inhibits NK cell activation pathways such as NKG2D, and dampens NK cell-mediated cytotoxicity towards malignantly transformed cells [10,89]. Despite its suppressive nature, Siglec-7 expression can define an active NK cell phenotype whereby the loss of Siglec-7 suggests NK cell dysfunction in primary hepatocellular carcinoma [90]. More recently, Siglec-7 expression was found to be transcriptionally regulated through DNA methylation/demethylation in NK cells [89]. Within the myeloid compartment, Siglec-7 in monocytes but not natural killer or T lymphocytes was reported to induce a pro-inflammatory response in the absence of sialic acid [91]. On the other hand, increased sialylation of pancreatic ductal adenocarcinoma can induce tumor-associated macrophage differentiation via Siglec-7 and Siglec-9 monocytes [92]. High levels of Siglec-7 in intratumoral macrophages were also associated with poor outcomes in vaccinated and metastatic colorectal cancer patients [93]. Hence, Siglec-7/-9-based CAR that bind tumor-associated Siglec-ligands is a new therapeutic effort for immunotherapy [94]. However, it is noteworthy that Siglec-7 ligands are also found in normal lung tissues [87], thus raising concerns in off-target cytotoxicity.

### 2.8. Siglec-8

Siglec-8, a CD33-related family member, is often found on human eosinophils, mast cells, and weakly on basophils [87]. Although it can exist as two spliced variants, Siglec-8 is described as the “long form” bearing two tyrosine cytoplasmic motifs and not its “short form” [95]. Siglec-8 is known to bind to a defined spectrum of ligands such as sialylated N-acetyllactosamines (LacNAcs), a shared ligand with galectins and 6’-sulfo-sialyl Lewis^x^ with tight specificity [96,97,98]. As a target in allergen-induced inflammation, Siglec-8 antibodies have recently emerged as treatment for eosinophilic gastritis and eosinophilic duodenitis [99], and are comprehensively reviewed elsewhere [100]. However, their role in cancer remains understudied. Nonetheless, high Siglec-8 expression was observed in luminal-like breast cancer and strongly correlated with tumor-associated epitope of MUC-1 expression. Similarly, elevated expression of Siglec-8 in tumors was correlated with poor OS in clear-cell renal cell carcinoma [101]. In contrast, reduced expression of intratumoral Siglec-8 predicts poor OS in patients with gastric cancer after surgical resection [101].

### 2.9. Siglec-9 (CD329)

Siglec-9 are highly homologous to Siglec-7 (V-set domain ~80% sequence identity), are MHC class I-independent inhibitory receptors, and are similarly found in myeloid cells, NK cells, and a subgroup of PD-1^+^ CD8^+^ T cells to resolve ongoing inflammation [102,103,104]. Siglec-9 also binds to a broader spectrum of sialoglycans, recognizes Sia-self-associated molecular patterns (Sia-SAMPs), and transduces Siglec-mediated cell apoptosis of leukocytes [105,106]. The protection of tumor cells can result from increased Sia-SAMP by engaging both Siglec-7 and Siglec-9 receptors to inhibit cell mediated tumor cell killing [13,107,108]. Moreover, Siglec-9 has also been reported in myeloid cell-mediated cancer progression via binding to MUC-1 with sialylated T-antigen (MUC-1 ST) in breast cancer and pancreatic ductal adenocarcinoma to modulate tumor-associated macrophage differentiation [92,109]. The enzyme responsible for attaching a sialic acid to T antigen in many types of cancer is ST3Gal1 [110]. Additionally, MUC-16 (CA125), which is primarily upregulated in ovarian cancer cells, binds to Siglec-9-positive leukocytes via α2,3-linked sialic acids [111]. Given the inhibitory nature of myeloid Siglec-7 and Siglec-9 receptors, the addition of soluble Siglec-9 was previously shown to provide antitumor benefit, and more recently the use of antibody blockade to favor antitumor response [112,113]. Indeed, the Siglec-9 receptor is a potential target as it is highly expressed in acute myeloid leukemia [114]. In addition, Siglec-9 blockade restored neutrophil-mediated killing of tumor cells [107]. These studies suggest that the Siglec-9 to MUC-1 ST/MUC-16 axis in tumor growth presents a potential inhibitory pathway for immunotherapy.

### 2.10. Siglec-10

Siglec-10 is an inhibitory ITIM-bearing a CD33-related Siglec receptor predominantly found on NK cells, B cells, monocyte-derived dendritic cells, and antigen-activated CD4^+^ T cells to repress immune reactivities [115,116,117]. Typical Siglec-10 ligands are heavily glycosylated molecules, such as CD24 and CD52, predominantly expressed in leukocytes [4,118] as well as α1-acid glycoprotein (AGP) found in hepatic plasma, extra-hepatic tissues, and immune cells [119,120]. Present studies between AGP and Siglec-10 remain unclear. On the other hand, several studies have reported that the CD24-human Siglec-10 axis (or murine Siglec-G homolog) recognizes endogenous DAMP to repress DAMP-mediated innate inflammatory responses [121,122,123]. Enhanced CD24 on tumors is also found to be associated with Siglec-10^+^ TAMs even in the absence of sialic acid [4]. This interaction promotes tumor evasion in ovarian cancer and triple-negative breast cancer [4]. Thus, blockade of Siglec-10^+^ TAMs has observed improved CD8^+^ T cell cytotoxicity and immunotherapy in patients with hepatocellular carcinoma (HCC) [124]. In addition, Siglec-10^+^ NK cells are found at higher frequency in human HCC tissues as well as elevated Siglec-10^+^ expression on T cells due to the presence of malignant extracellular vesicles, promoting tumor progression [115,125]. Besides the CD24 ligand, high CD52 expression is found on activated CD4^+^ T cells, which do not normally express the Siglec-10 receptor [117]. Secreted and soluble CD52 is known to bind to the Siglec-10 receptor found on activated T cells via the DAMP molecule, HMGB1, to suppress T cell function [116,117]. Notably, HMGB1 can interact with CD21 or CD52 to form a trimolecular complex with Siglec-10 [116,122]. Soluble CD52 can also inhibit innate cells such as macrophages, monocytes, and dendritic cells [126]. Although CD52 is highly expressed in chemoresistant EVl1^+^AML with poor outcomes, it is a prognostic marker of breast cancer with good OS in patients [127,128]. 

### 2.11. Siglec-11 and Siglec-16 Pairs

Paired Siglecs share similar ligand-binding regions and cellular expression but opposite signaling pathways. Besides the Siglec-5/-14 receptor pair, inhibitory Siglec-11 is paired with activating Siglec-16 receptor (first two Ig-like domains: 99% of sequence identity), and expressed on macrophages/microglia of the CNS and fibroblasts from ovary [129,130,131]. Siglec-11 has a cytoplasmic ITIM motif, but Siglec-16 does not have a signaling motif. Instead, Siglec-16 binds to an adaptor molecule, DAP12, containing the ITAM to trigger an activating function [132]. Although several studies have reported Siglec-11 or -16 in mediating microglial activities [133,134,135], there are fewer studies related to various cancers. While Siglec-11 expression on ovarian stroma was previously known [129], it is not prognostic for ovarian cancer. More recently, a study showed that dexamethasone/temozolomide-treated glioma displayed higher affinities towards inhibitory Siglec-5 and Siglec-11 [136]. However, it remains unknown if both glioma and accumulating mast cells express the same Siglec-11 ligands [137]. In terms of our understanding of Siglec-11 and Siglec-16 in cancer, it still remains unclear.

### 2.12. Siglec-12 (Siglec-XII)

Siglec-12 is found in some macrophages and as lowly expressed in human epithelium, but is not a bona fide Siglec as a highly conserved Arg residue is lost during human evolution [138]. This results in a non-Sia binding Siglec, inefficient ITIM signaling, and its inability to distinguish between itself and foreign molecules. Nonetheless, Siglec-XII is able to recruit SHP2-related oncogenic pathways. An elevated Siglec-XII level is also associated with advanced colorectal cancer and poor OS in patients [139].

### 2.13. Siglec-13

The Siglec-13 gene was deleted during an Alu-mediated recombination event in humans but still present in chimpanzees and baboons [140].

### 2.14. Siglec-15

Siglec-15 is normally expressed on a subset of myeloid cells that expresses dendritic cell-specific intercellular adhesion molecule-3-grabbing non-integrin (DC-SIGN) but is also broadly upregulated on cancer cells and tumor-infiltrating myeloid cells [141,142]. Like Siglecs-14 and -16, Siglec-15 has an extracellular domain consisting of two Ig-like domains and lacks a cytoplasmic tail. Unlike most Siglecs, which are inhibitory receptors, it associates with adapter proteins DAP10 and DAP12 via a transmembrane Lys^274^ residue for signal activation [143]. However, it is a critical immune suppressor of T cells [142]. In the tumor microenvironment, Siglec-15 predominantly binds to tumoral sialyl-Tn structure (Sia-α2,6 linked to GalNAc α–O-Ser/Thr), which is associated with poor prognosis in patients with gastric, colorectal, or ovarian cancers [144,145], as well as α2,3-linked sialic acid [146]. Additionally, positive Siglec-15 expression on tumor surfaces is often correlated with poor OS in lung cancer patients [147]. Overexpression of Siglec-15 on tumor associated macrophages can contribute to tumor immunosuppression via production of anti-inflammatory transforming growth factor-β along the DAP12-Syk pathway but can be downregulated by IFN-γ [142,148]. This is unlike PD-L1-expressing macrophages, which do not inhibit T cell responses but only evade T cell killing [149]. At the mRNA level, high Siglec-15 mRNA was associated with worse OS in pancreatic ductal adenocarcinoma, sarcoma, and kidney renal clear cell carcinoma but is favorable in breast cancer, head and neck squamous cell carcinoma, thyroid carcinoma, and uterine cropus endometrial carcinoma; these differences may be due to prognoses [150]. Additionally, sialyl-Tn is not detectably found on certain cancer cells but can still recognize the Siglec-15 receptor via other sialylated structures [151]. Additionally, Fudaba and coworkers recently showed that Siglec-15-positive peritumoral and non-intratumoral macrophages correlated with better OS in primary central nervous system lymphoma patients. More recently, high Siglec-15 was also found to promote osteosarcoma progression via the activation of the DUSP1/MAPK signaling pathway [152], and hepatocellular carcinoma migration via CD44 interaction, which prevented lysosomal degradation [153]. As Siglec-15 is mutually exclusive to PD-L1, it is now emerging as a novel immune inhibitor for anti-PD-1/PD-L1 resistant patients [154,155]. 

### 2.15. Siglec-17

Siglec-17 is an inhibitory molecule found in chimpanzees. It is non-functional in humans due to a deletion in the open reading frame of gene *SIGLEC17*, but its pseudogene mRNA is still detected in human natural killer cells [140].

### 2.16. Siglec-Related Downstream Immune Signaling

Upon sialic acid binding to Siglec, a cascade of downstream signaling pathways modulate the immune responses (Figure 3). The majority of Siglecs are inhibitory (Siglec-2 until Siglec-11, excluding Siglec-4), and bear ITIM or ITIM-like motifs in the cytoplasmic tails. These ITIM/ ITIM-like motifs can be phosphorylated by the Src family of protein tyrosine kinases, which later recruit protein tyrosine phosphatases SHP-1 and SHP-2. Both SHP-1 and SHP-2 play important inhibitory roles during lymphocyte activation by dephosphorylating key signaling molecules and antagonizing tyrosine kinases [156]. Additionally, SHP-1 and SHP-2 can inhibit ITIM-dependent endocytosis of Siglecs, whose mechanism remains elusive [157]. Mechanistically, SHP-1 can dephosphorylate a number of kinases such as Lyn, SyK, Lck-56, and ZAP-70 [158,159,160,161], and is a known negative regulator of immune cell signaling of Janus kinase (JAK)/ signal-transducer and activator of transcription (STAT) and toll-like receptor (TLR) signaling [161,162,163]. Furthermore, the lack of SHP-1 can prolong the activation of pro-inflammatory transcription factor nuclear factor-κB (NFκB) [164], which can increase the production of cytokines such as TNF and IL-1 and inflammasome regulation [165]. Similarly, SHP-1 deletion in T and B cells in mice can cause TCR-mediated proliferation and even autoimmunity/autoinflammation, respectively [166,167], and hence SHP-1 inhibits cell signaling.

Counterintuitively, SHP-2 is a positive effector of receptor tyrosine kinase (RTK) signaling by inactivating Sprouty (Spry), a RTK inhibitor (Figure 3) [168], as well as promoting Ras/Mitogen-activated protein kinase (MAPK), JAK/STAT, extracellular signal-regulated kinase (ERK) and PI3K/Akt cascades [169,170]. SHP-2 is an oncogene that regulates both immune cell and tumor signaling, and its dysregulation is associated with the progression of leukemias and gastric and breast cancers [171,172,173]. Furthermore, studies have shown that SHP-2 maintains Ras/MAPK activation whereby MAPK-hyperactivating mutations of SHP-2 have led to human diseases [174,175]. In addition, CD33 suppressor of cytokine signaling 3 (SOCS3) was previously reported to compete with SHP-1/-2 for phosphorylated ITIMs in Siglec-3 (and possibly CD33-related Siglecs), leading to CD33 proteasomal degradation [176]. As CD33 can repress cytokine signaling [177], the loss of CD33 and gain in SOCS3 can function to enhance cytokine responses.

In contrast, Siglec-14, Siglec-15, and Siglec-16 do not have ITIM or ITIM-like motifs. These are activating Siglecs that associate with DAP12, which recruits SyK (Figure 3) [178]. The DAP12-bound SyK complex can recruit proximal signaling molecules such as phosphatidylinositol-3-OH kinase (P13K) and subsequently trigger cellular activation [141,179]. Furthermore, Siglec-14 can activate the inflammatory mitogen-activated protein kinase (MAPK) pathway in neutrophils and enhance NLRP3-inflammasome activation in macrophages [66,74]. Hence ITAM-containing DAP-associated Siglecs can activate the immune system. Further readings on the dichotomy between ITAM and ITIM signaling pathways are available elsewhere [10,180,181].

## 3. Siglec-Directed Strategies to Counteract Tumor Immune Evasion

Tumor cells have evolved to exploit hypersialylation to trick the immune system into recognizing them as “normal self”, resulting in attenuated immune cell activity (Figure 4) [189]. Hence, to counteract tumor immune evasion, sialic acid binding Siglec-receptors are potential targets [190]. Here, we will describe two major themes in current targeted cancer therapies against the restricted expression of Siglecs based on (1) antibody design (2) synthetic sialosides (Figure 5). 

### 3.1. Anti-Siglec Based Immunotherapies

Despite advances in standard chemotherapy and allogenic hematopoietic stem cell transplantation (alloHSCT), their associated non-specific toxicities and high mortality rates present obstacles for achieving durable remission in hematological malignancies. Monoclonal antibodies (mAbs) targeting Siglecs are an emerging alternative modality for passive immunotherapy to deplete tumor cells [191,192]. Traditionally, most mAbs mediate cell death via complement-dependent cytotoxicity (CDC) or antibody-dependent cellular cytotoxicity (ADCC) [193]. However, recent strategies target the immune effector cells to either enhance (bispecific antibodies) or re-direct (CAR-based cell transfer) their antitumor capabilities, and are discussed here [194].

#### 3.1.1. Naked Antibodies for Tumor Cell Lysis

Traditional unlabeled (also referred to as “naked”) mAbs can target specific Siglec-positive cells to induce apoptosis. The expression of certain Siglec members, such as the B-cell surface-restricted Siglec-2 (CD22), has garnered growing interest for cell-directed antitumor therapies. Moreover, CD22 expression increases by 60% to 80% in B-cell malignancies [195], making it an ideal candidate for treating various B-cell lymphomas [191]. Previously, a humanized IgG1 monoclonal targeting CD22 (Epratuzumab) has undergone extensive evaluation as treatment for various subtypes of non-Hodgkin’s lymphoma (NHL) (Figure 5A). Early phase I/II clinical trials have shown that Epratuzumab as a single agent is well tolerated, and can induce tumor regressions in patients with indolent follicular lymphoma (FL) and aggressive diffuse large B-cell lymphoma (DLBCL) [196,197]. While unlabeled Epratuzumab showed only modest levels of ADCC, its combination with FDA-approved rituximab targeting B-cell antigen CD20, showed enhanced cytotoxicity in phase II studies in patients with relapsed/refractory (r/r) NHL. 

Next, myeloid-cell-expressing inhibitory Siglec-7 and Siglec-9 play a crucial role in T-cell activation and antitumor response. Recent in vitro and in vivo studies have revealed that Siglec-7 and Siglec-9 promote tumor progression in humanized knockout murine models by supporting the immunosuppressive tumor microenvironment [113]. Moreover, monoclonal antibodies targeting Siglec-7 (clones 1E8) and Siglec-9 (mAbA) have led to tumor burden reduction in vivo mouse model (Figure 5B) [113]. These studies demonstrated the potential of targeting immunomodulatory Siglec receptors as a means to increase antitumor immunity with reduced tumor burden.

#### 3.1.2. Antibody Drug Conjugates (ADC) for Specific, Potent Tumor Targeting

Despite improvements in the disease outcome in patients with B-cell malignancies using single-agent or tandem mAb-targeted therapy, most fail to achieve sufficient cytotoxicity and/or complete remission. Alternative therapies are needed for patients with relapse or refractory (r/r) disease (after failed standard chemotherapies or prior immunotherapies) and elderly or frail patients who are unsuitable for aggressive chemotherapeutics. To overcome these hurdles, another class of targeted therapy, tripartite antibody-drug conjugates (ADCs), was developed. ADCs harness specific mAb as a vehicle to deliver potent cytotoxic drugs to tumor cells with minimized off-target toxicity. Several Siglec-directing ADCs have been approved by the FDA for clinical use in haematological neoplasms including Inotuzumab ozogamicin (InO; Besponsa®; Pfizer; CMC-544) and Gemtuzumab ozogamicin (GO; Mylotarg; CMA-676; Wyeth Laboratories, Philadelphia, PA, USA), targeting CD22 and CD33 respectively (Figure 5C) [198,199]. Both ADCs are humanized IgG4 monoclonal’s conjugated to N-acetyl-ɣ-calicheamicin dimethylhydrazide, a potent DNA-damaging antibiotic. Mylotarg was the first ADC approved by the FDA in May 2000 for treating relapsed AML but was later withdrawn due to early fatalities in newly diagnosed AML cases [200]. However, Mylotarg was re-approved by the FDA in 2017 for the treatment of newly diagnosed, relapsed, and refractory AML patients. The approval was based on findings from critical research that indicated the advantages of fractionated Mylotarg treatment outweighed the adverse risks [201]. 

Motivated by the success of Mylotarg, a similar immunoconjugate targeting CD22, Inotuzumab ozogamicin, was recently approved as monotherapy for adults with r/r B-cell precursor CD22-positive acute lymphoblastic leukemia (ALL). InO’s success was partly contributed to by CD22’s ability to undergo constitutive endocytosis into B-cells following antibody ligation (for details refer to the excellent review [202]). InO demonstrated a significantly higher complete remission rate (80.7% vs 29.4%) with a higher percentage having results below the minimum residual disease (MRD) threshold (78.4% vs. 28.1%), progression-free survival (5 months vs 1.8 months), and OS (7.7 vs 6.7 months) with 11% suffering from treatment-emergent hepatotoxicity [203]. InO has proven efficacy as a bridge to transplant therapy in Philadelphia negative and positive (Ph^−^, Ph^+^) patients suffering from CD22 positive relapsed extramedullary-ALL [204]. Similarly, promising results have been reported in pediatric ALL patients (phase II) and r/r B-cell NHL patients (including FL, DLBCL, and MCL) [205,206,207,208]. Despite these, phase III trials were stopped following an interim evaluation suggesting that the study would not reach its end point [209].

#### 3.1.3. Immunotoxins: Antibody-Derived Protein Toxin Conjugates

Immunotoxins (RITs) are recombinant proteins which express a variable fragment (Fv) of monoclonal antibody fused to either bacterial or animal toxins. Contrary to drug conjugates, immunotoxins’ cytocidal activity relies on the targeting antigen’s specificity as well as the toxin moiety’s activity to induce cell death, and with reduced dose-limiting toxicity [210]. A CD22-targeting immunotoxin, Moxetumomab pasudotox (Moxe, CAT-8015/HA22), was developed for treating hairy cell leukemia (HCL) (Figure 5D). Moxe contains a single chain Fv of anti-CD22 fused to a Pseudomonas endotoxin A (PE38) moiety. Upon internalization and translocation to the cytosol, the drug induces cell death via blocking protein synthesis independent of ADCC or CDC activity (Figure 5D). Based on promising results from phase I trials (NCT00462189) in pre-treated r/r HCL patients, phase III studies were conducted to determine the complete response with maintenance of hematologic remission (HR). The response rate was 41% (33/80) with an MRD negativity rate of 82% (27/33) while 75% of the enrolled patients (60 of 80) reached objective response with an HR rate of 80% (64 of 80) [211]. These led to FDA approval of Moxe for treatment of patients with relapsed or refractory HCL in 2018. Currently, clinical trials using Moxe in combination with rituximab (anti-CD20 mAb) are underway for treatment of pediatric patients with r/r HCL and HCLv (variant of HCL) [210].

#### 3.1.4. Bispecific Antibodies Targeting T Cells

Bispecific antibodies (bsAbs) are a fusion of two different antibodies corresponding to two different antigens or distinct epitopes of a single antigen in various formats. One particular format targets the cytotoxic potential of immune effector cells (IECs) by bridging an activating receptor on IEC (e.g., CD3ε on T cells) and an antigen (e.g., CD22) on malignant cells for subsequent tumor cell lysis [212,213]. Superior to unconjugated mAbs, bispecific antibodies exhibit greater efficacy in treating hematological malignancies. Several bsAbs (AMG330, AMV564, and AMG673) targeting CD33^+^ AML are being evaluated in preclinical and early clinical trials (NCT02520427, NCT03144245, NCT03224819) to determine their efficacy as immunotherapeutic agents.

AMG330, a human BiTE (Bispecific T-cell engager) directed against CD33/CD3 led to T-cell expansion (Figure 5E), differentiation, proliferation, and produced a potent T cell cytotoxic response against primary AML blasts, even at low effector to target (E:T) ratios, validating it as a candidate for targeting AML [214]. The study established a correlation between CD33 expression levels with disease characteristics in patients with newly diagnosed AML. Low CD33 expression correlated with complex karyotypes while *NPMI* mutations were associated with high CD33 expression. Clinical trials are ongoing (NCT02520427) to evaluate the safety and tolerability of CD33/CD3 bispecific antibody in 55 patients with r/r AML. Although preliminary results from this study indicate that AMG330 showed potent anti-leukemic activity, their rapid renal drug clearance required continuous intravenous infusion (CIV) for up to several weeks [215]. 

To reduce renal clearance and subsequently reduce CIV frequency, two approaches are introduced. In the first approach, AMV564, a tetravalent bispecific tandem diabodies (TandAbs) with a molecular weight (106 kDa) exceeding the renal clearance threshold showed a longer half-life than AMG330 [51]. CD33/CD3 TandAbs (T564) also demonstrated cytotoxic activity independent of CD33 cell surface density and disease characteristics in r/r AML patient-derived samples and human CD33^+^ AML cell lines [216]. Preliminary results from an ongoing phase I study (NCT03144245) confirmed its anti-leukemic activity through T-cell engagement in patients with r/r AML. In the second approach, AMG673 is an anti-CD33/anti-CD3 half-life extended (HLE) BiTE construct. The HLA BiTE construct bears the N-terminus of single chain IgG Fc domain, which prolonged its half-life in r/r AML patients previously treated using other anti-AML treatments [217]. Despite extensive efforts to enhance the cytotoxic activity of T cells against AML blasts, the T-cell engaging bispecific still showed limited efficacy and severe adverse events post treatment, leading to early termination [218]. 

#### 3.1.5. Bi- or Tri-Specific Antibodies Targeting NK Cells

With a better understanding of the cytotoxic potential of NK cells in cancer therapy, tumor-specific NK cell-mediated ADCC using novel bi- and tri-specific killer engager antibodies (BiKE, TriKE) were generated [219,220]. Through the CD16 antigen, bispecific killer engager-containing anti-CD16 and anti-CD33 scFvs triggered enhanced NK cell cytotoxicity against CD33^+^ HL60 cell line and *de novo* AML cells independent of the E:T ratio [219]. Additionally, NK cell effector functions against *de novo* CD33^+^ AML cells were enhanced in the presence of the ADAM17 inhibitor, which inhibited CD16 shedding [219]. The specific enhancement of BiKE-mediated NK cell function was also observed in refractory AML blasts [219]. The efficacy of CD33/CD16 BiKE to induce NK cell function in myelodysplastic syndrome (MDS) cases showed significant NK cell degranulation and IFNγ and TNFα production in MDS patients [221]. The small-sized BiKE (60 kDa) construct also aided effective tissue penetration to reach the bone marrow where the tumor cells resided but increased susceptibility to renal clearance. Subsequently, the same group introduced the TriKE construct to improve the in vivo expansion of NK cells; a third scFv for IL-15 was introduced as a linker between CD16 and CD33 scFvs to enhance NK cell proliferation with antitumor function [220]. 

#### 3.1.6. CD19/CD22-Directed CAR T-Cell Therapy

Adoptive cell transfer (ACT) is a rapidly emerging form of immunotherapy which utilizes the patient’s own immune cells (allogenic) to detect and destroy malignant tumor cells. One of the ACT techniques, CAR-T cell-based treatment, is described as “a living drug" since it involves genetically modified T cells to express synthetic receptors called CARs on their surface (Figure 5F). CARs harbor a fragment of synthetic antibodies (scFv) which target specific tumor-cell antigens, and rely on stimulatory signals conveyed by an intracellular signaling module e.g., TCR CD3ζ chain for T-cell activation and enhanced immune response [222,223]. In recent years, a newer generation of CAR-Ts has been developed with increased T-cell proliferation and longer half-life in circulation post infusion [224]. The key to successful CAR T-based therapies is the choice of tumor antigen to be safe, effective, and specific. CD19-directed CAR-T therapy for adult and pediatric B-ALL failed to achieve CR due to shedding of the CD19 antigen. To address this issue, an alternative CD22-directed CAR-T therapy reported CR in 70% of patients with an OS rate of 1 year post therapy [225]. However, a drawback was the need for subsequent transplantation (allo-HSCT) to maintain durable remission in patients with relapse [225]. Combination therapy using CD19- and CD22-CAR-T in adult and pediatric B-ALL patients significantly enhanced the median OS (88.5% for 12 months) and progression-free survival (survival rate 67.5% for 18 months) rate and duration [226]. Similarly, dual CD19/CD22-targeted CAR T cells achieved complete remission with low-to-moderate grade treatable CRS in patients with refractory DLBCL [227].

#### 3.1.7. CD33-Directed CAR T-Cell Therapy

Building on the success of CAR-T therapy in the treatment of B-cell malignancies, efforts have been made towards developing similar CAR-T strategies to improve clinical outcomes in AML patients. Despite the preclinical success of CD33-directed CAR-T-therapy in inducing anti-leukemic effects, in vivo targeting of AML blasts is difficult due to the overlapping expression of AML associated-antigens on progenitor and mature hematopoietic cells of the myeloid lineage. AEs such as hepatic damage and an immunosuppressive milieu that hindered T cell-mediated killing were also unfavorable. To circumvent these challenges, Cartellieri et al. developed the UniCAR platform, which consists of a tumor-specific inactive CAR and a targeting module (TM) that works as an on-off switch for CAR-T antitumor activity [228,229]. The dual specific anti-CD123-CD33 TM allowed TM-restricted redirection of T-cell cytotoxicity towards tumor cells presenting both epitopes. UniCAR improved the clinical usability of traditional CAR for tumor cell targeting by minimizing on-target off-site events and preventing tumor escape [228]. 

### 3.2. Sialic-Acid Ligands—Triumph over Siglec Immunosuppression

Over the years, specific, high-affinity sialosides-ligands have been developed to target tumor-related Siglecs as an alternative immunomodulatory strategy to halt cancer progression. Natural sialic acid-containing glycans serve as important regulators of the Siglec checkpoint in immune cell responses, involving cytoplasmic regulatory motifs (ITIM/ITAM), Siglec microdomain localization, and cis/ trans interaction via sialoglycans as ligands [9,230,231]. However, individual sialoglycoconjugates-Siglec interactions have weak intrinsic affinity (high μM to low mM range). Nature overcomes this by displaying multiple copies of the cis ligands (affinity ∼50–100 mM) to microdomain-clustered Siglecs on the same cell surface (“glycocluster” effect), as exemplified by the well-studied CD22 receptor [232]. Consequently, these cooperative interactions resulted in enhanced interaction avidity.

#### 3.2.1. Designing High Affinity Sialic-Acid Ligands

The low affinities of natural Siglec ligand binding posed a challenge in synthetic sialoside ligands for modulating Siglec function. Kelm et al. revolutionized this field by designing the first high-affinity synthetic analogues targeting CD22 derived from its natural sialic acid scaffold [233]. Specifically, a methyl group (9-amino-Neu5AcAMe) at C9, or halogenated acetyl containing three fluor atoms at C5 (Neu5F3AcAMe), greatly enhanced interaction affinity due to extra hydrogen bonding and lipophilic interactions between the sialoside and CD22 [234]. Additionally, to compete with natural multivalent Siglec binding and clustering, multivalent scaffolds presenting sialosides such as polyacrylamide and PAA (1000-kDa BPC-NeuAc-LN-PAA) have effectively outcompeted the cis ligands and bound to CD22 positive B-cells [232,235]. 

#### 3.2.2. Structural-Guided Sialoside Design

Advances in the structural information on Siglecs further aided in the rational design of high-affinity sialic acid mimetics (SAMs). The crystal structure of Siglec-1’s (sialoadhesin) V-set domain bound to its endogenous ligand 3’sialyllactose (NeuAc(α2,6)Lac) revealed the minimal sialic acid recognition site organized by three conserved Siglec residues, W2 (strand A), R97 (strand F), and W106 (strand G) within the ligand-binding domain [236]. Additionally, a homology model for CD22 was used to design numerous high-affinity CD22 ligands, including a trivalent Tris-based cluster of Neu5Ac-4-nitrobenzoyl-Glc with over 300-fold higher affinity compared with monovalent NeuAc(α2,6)Lac [237]. Another synthetic human CD22 inhibitor-ligand, BPC-Neu5Ac (biphenyl-4-carbonyl)-amino-9-deoxy-Neu5Ac), with over 200-fold affinity for hCD22 compared with Me-Neu5Ac was also developed [237,238]. Finally, the co-crystal structure of sialoadhesin with BPC-Neu5Ac revealed a hydrophobic pocket which accommodated the biphenyl substituent to explain its high Siglec-binding affinity [239]. 

#### 3.2.3. Sialoadhesin/CD22-Directed Liposomes in Cancer Therapy

Over the years, the roles of glycans in cancer have become valuable in developing antitumor therapies. SAMs are synthetic ligands which mimic the oligoantennary oligosaccharides found on endogenous glycoconjugates, and have shown potential as targeting moiety to selectively bind Siglec-expressing tumor cells [240]. Various platforms including liposomes have also been explored as lipid-based vehicles for these glycans. For example, liposomal nanoparticles (NPs) decorated with high-affinity CD22’s glycan ligands have been used to deliver the cytotoxic drug, doxorubicin (Dox), to B-lymphoma cells (Figure 5G) [241]. The CD22-targeted liposomes exhibited 33-fold increased efficacy (IC50 = 1.6 μM) in killing cells compared with naked liposomes (IC50 = 53 μM), and significantly extended survival in a xenograft model of human B-lymphoma [241]. Hence the Dox-loaded liposomal NPs is able to target malignant B cells from blood samples of patients with hairy cell leukemia (HCL), chronic lymphocytic leukemia (CLL), and splenic marginal zone lymphoma (MZL) [242,243]. Its mode of action involves the dissociation of the Dox-loaded liposomal NPs complex within the acidic environment of endosome, subsequent translocation to the cytosol, and inducing apoptosis. More importantly, the recycling of CD22 acts as a shuttle to transport new cargo back into the cell to greatly enhance cytotoxic activity [244]. Sialic acid-modified liposomal-carriers have also been used to deliver anti-cancer drugs such as Pixantrone (Pix) and Epirubicin (EPI) to Siglec-1-expressing TAMs [245,246]. The liposomal carriers target and eradicate both TAMs and peripheral blood monocytes via internalization of loaded liposomes, leading to the depletion of TAMs in the tumor microenvironment [247]. Additionally, the antitumor activity of Epirubicin-loaded liposomes decorated with sialic acid-cholesterol conjugate was found superior to Pix-loaded liposomes modified with sialic acid–octadecylamine conjugate [246]. 

## 4. Discussion

Siglecs are protein receptors in glyco-immunology, which are fundamental in understanding the biology of various cancer progression in patients. Considering that hypersialylation and tumor immune evasion are in a close relationship with poorer OS in patients, more work on other understudied Siglecs, soluble Siglecs, and their tumor-associated sialic acid ligands are needed. For example, the roles of inhibitory Siglec-11 and Siglec-XII in myeloids still remain an enigma in many cancers. Additionally, new findings on the Siglec-10-CD24 axis in TAMs revealed possible sialic-acid independent pathways in the tumor microenvironment [4]. Here, we also highlighted different strategies that primarily target Siglec-1, Siglec-2, Siglec-3, Siglec-7, and Siglec-9 expressions on malignant cells. Various anti-Siglec formats such as naked, ADCs, immunotoxins, bispecific, and CAR-T strategies that disrupt Siglec-ligand interactions and alter the immune response have been developed. These blocking antibodies can deliver antagonizing or agonizing signals, leading to either amplification or attenuation of the antitumor responses. Additionally, similar to the ADC properties of anti-Siglecs, SAMs have exploited the endocytic properties of Siglecs to deliver potent drugs or antigens to kill tumor cells or induce antitumor immunity, respectively. We also highlighted the natural low-binding affinities of sialic-acid ligands, and discussed various strategies to compete and enhance Siglec-ligand binding. These include the use of multivalent scaffolds such as polyacrylamide and liposomal NPs as well as better SAM designs. However, systemic toxicity associated with SA-modified liposome drug conjugates can be a problem. To eliminate any systemic toxicity, glyco-modified NK cells were recently designed to specifically target the expression of NK-activation receptor (NKG2D) restricted on malignant cells, and the expression of E-selectin ligand to traffic to the bone-marrow-residing tumor cells expressing CD22 (Figure 5H) [248]. Thus, glyco-engineering on the NK cells may provide an attractive scaffold for multivalent ligand display and tumor cell lysis while leaving the healthy B-cells untouched.

## 5. Conclusions

Siglecs are becoming widely recognized as potential targets to modulate the effector cell-signaling responses for therapeutic applications, such as in hematological malignancies [19,249]. Many more studies conducted in the future will definitely improve our understanding in Siglecs and their roles in cancer biology. Considering that Siglecs and sialic acid structures are in constant interaction within the tumor microenvironment, high-content multiplexed technologies, such as single-cell RNA sequencing and mass cytometry, are recommended to dissect the massive heterogeneity at the glyco-immunology interface. Incorporating advances in protein modeling, such as AlphaFold, will also guide better SAM design to develop better competitive inhibitors, elucidate new binding pockets, and provide clues required to understand cryptic sialic acid-independent interactions. Currently, establishing a comprehensive understanding of all Siglecs has not been completed, but with the maturation of methods and new discoveries, it may eventually lead to more effective treatment and prevention against cancer.

## Figures and Tables

**Figure 1 biology-10-01178-f001:**
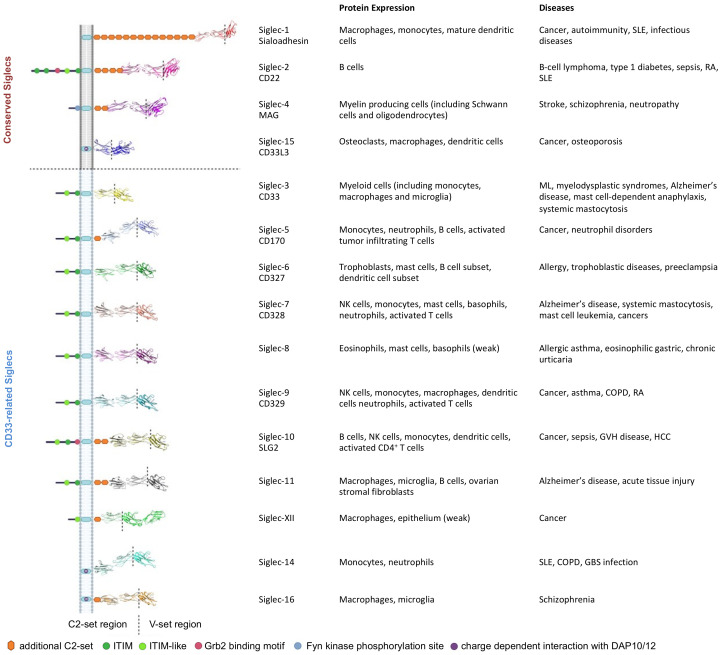
Human sialic-acid binding Siglec family members associated with various diseases. COPD: chronic obstructive pulmonary disease. GBS: Group B Streptococcus. GVH: graft-vs-host. HCC: hepatocellular carcinoma. ML: Myeloid leukemia. RA: rheumatoid arthritis. SLE: Systemic lupus erythematosus. Siglec-12 lacks a highly conserved Arg residue for sialic-acid binding and is denoted as Siglec-XII. The Siglec structures were built without the secretion leader using AlphaFold [22].

**Figure 2 biology-10-01178-f002:**
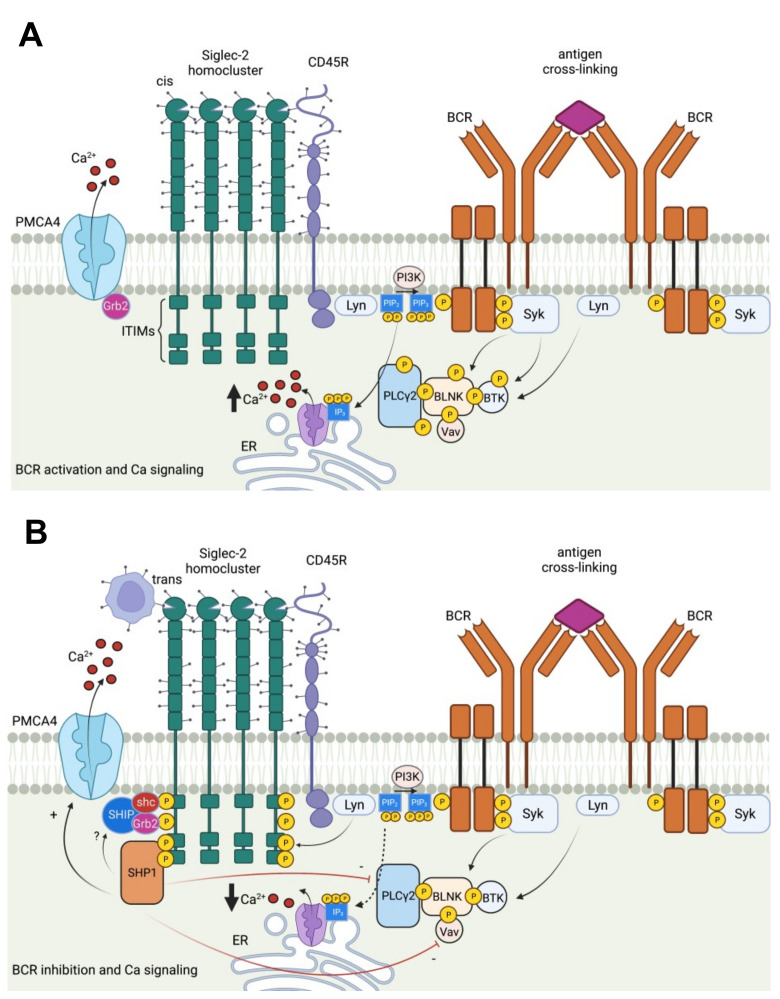
Activating and inhibitory pathways in Siglec-2 (CD22) as targets in cancer. (**A**) Antigen cross-linking of BCR results in an increase in cytosolic calcium in the B cells. Upon BCR stimulation, phosphorylation and activation of different BCR signaling complexes occur. Antigen binding is followed by the phosphorylation of ITAMs of Ig complex and PLCγ2 mediated production of signaling molecule, inositol-1,4,5-triphosphate (IP3). IP3 binds to IP3 receptor on the endoplasmic reticulum (ER) to release Ca^2+^ into the cytosol. Spleen tyrosine kinase (Syk) leads to increased phosphorylation of BTK-BLNK-PLCγ2-VaV complex in the calcium cascade, which releases more Ca^2+^ out of the ER. (**B**) Following BCR activation, cis or trans Siglec-2 recruitment inhibits BCR signaling and is phosphorylated by Lyn. There are six tyrosines in the intracellular tail of Siglec-2, out of which three are in the ITIMs. Two phosphorylated ITIMs can bind to tyrosine phosphatase SHP-1, which decreases the release of Ca^2+^ at the ER. Through the formation of quaternary SHIP-Grb2-Shc-CD22 complex, Ca^2+^ is further released through plasma membrane Ca^2+^ ATPase (PMCA, a calcium pump), to terminate calcium responses in the B cells after antigen stimulation.

**Figure 3 biology-10-01178-f003:**
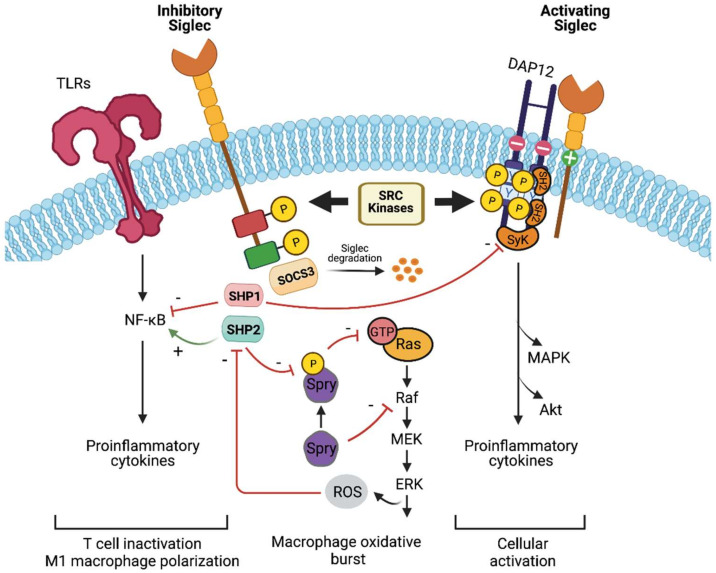
Siglec signaling and immune modulatory effect. Upon sialic acid binding, SRC kinases can phosphorylate tyrosine residues found in the cytosolic ITIM/ ITAMs. Within the inhibitory Siglecs, SHP-2 is recruited to the phosphotyrosines and can inhibit Sprouty (Spry). The inhibition of Spry promotes the activation of Ras-ERK signaling and subsequent ERK and Akt-dependent production of reactive oxygen species (ROS) [182,183]. ROS in sepsis can inactivate SHP-2, resulting in endothelial activation [184]. Additionally, SHP-2 can activate the NK-κB pathway, leading to inflammation and cellular immune responses [185]. On the other hand, SHP-1 inhibits the NK-κB pathway and controls cytokine production through the JAK-STAT pathway [164,185]. SHP-1 is associated with T cell inactivation, whereas SHP-2 is associated with the activation of M1 macrophage phenotype [166,186]. Suppressor of cytokine signaling 3 (SOCS3) competes with both SHP-1/-2 for phosphorylated ITIMs, thereby inhibiting signaling and resulting in accelerated proteosomal degradation of both Siglec-3 and SOCS3 [176]. Next, within the inhibitory Siglecs, DAP12 associates with the positively charged residue found in the transmembrane of inhibitory Siglecs. Subsequently, Syk bearing tandem SH2 domains binds to the phosphorylated ITAM-containing DAP12 [187], and results in the downstream generation of second messenger, elevated intracellular calcium, and the activation of MAPK/Akt pathways. Notably, Syk can be inhibited by SHP-1 [188], and thus modulate the activating and inhibitory signaling of Siglecs.

**Figure 4 biology-10-01178-f004:**
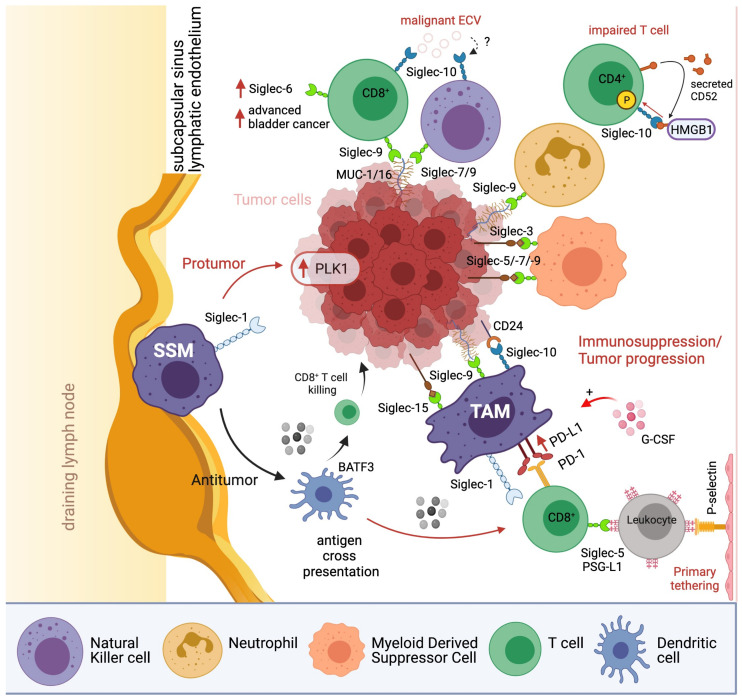
Overview of inhibitory mechanisms of the Siglec^+^ immune cells in the tumor microenviroment. Protumor pathways are marked with red arrows and antitumor pathways are marked with black arrows. BATF3: Basic Leucine Zipper ATF-Like Transcription Factor 3, ECV: Extracellular vesicles, HMGB1: High-mobility group box-1, SSM: Sinusoidal macrophages, TAM: tumor-associated macrophages, PD-1: Programmed cell death protein 1, PD-L1: Programmed cell death ligand 1, PLK1: Polo-like kinase 1, PSG-L1: P-selectin glycoprotein ligand-1. SSM: subcapsular sinus macrophages. TAM: tumor-associated macrophages.

**Figure 5 biology-10-01178-f005:**
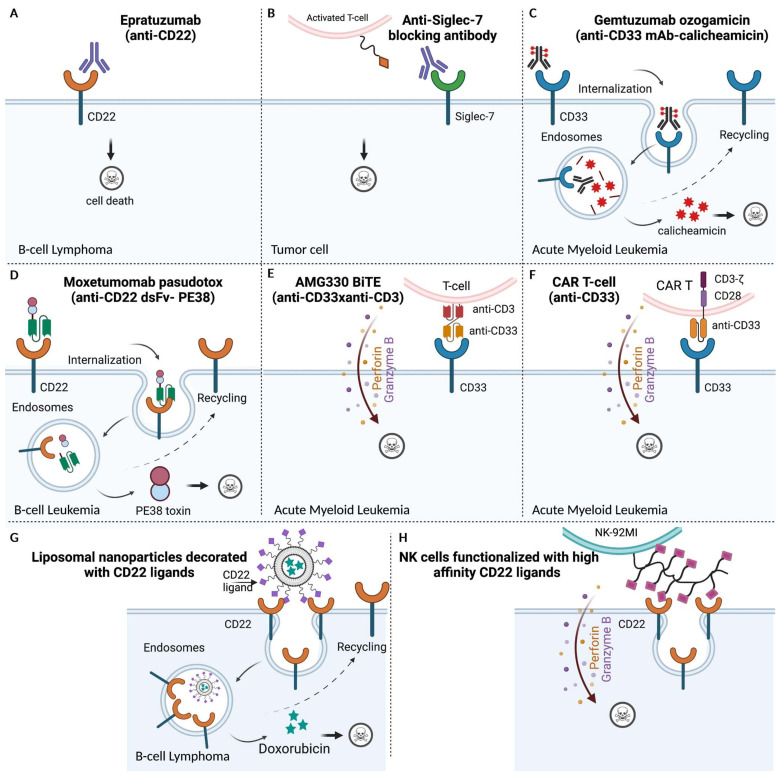
Siglec-directed strategies as therapeutics. The three modalities used to target Siglec receptors: (**A**–**D**) Anti-Siglec antibodies, (**E**,**F**) redirection of cytotoxic immune cells, and (**G**,**H**) high-affinity, multivalent Siglec-ligands. Antibodies mediate anti-tumor effects via different mechanisms: target cell depletion via (**A**) Siglec-specific monoclonal antibodies. Epratuzumab, a humanized mAb directed against the B-cell restricted antigen, CD22/ Siglec-2 on B-cell lymphomas, and (**B**) anti-Siglec-7 enhances T cell cytotoxicity by delivery of cytotoxic payloads to cancer cells via (**C**) Anti-Siglec antibodies conjugated to cytotoxic drugs such as Gemtuzumab ozogamicin (GO) that exploit the endocytic property of CD33 receptors for targeted delivery of cytotoxic drugs to CD33^+^ AML blasts; (**D**) Moxetumomab pasudotox, an immunotoxin derived from the scFv fragment of anti-CD22, fused to bacterial endotoxin, PE38, for the treatment of B-cell leukemia; Siglec-sialoglycan blockades using antibodies similar to (**A**,**E**), AMG330 BiTE enhances T cell cytotoxicity or (**F**) uses CAR-T cells to retarget the immune response towards tumor cells. Other strategies also include multivalent drug delivery platforms such as (**G**) liposomes and (**H**) glycoengineered cytotoxic NK cells modified with high-affinity Siglec ligands for selective targeting of tumor cells.

## Data Availability

Not applicable.

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
