# Peer review of "Siglecs as Therapeutic Targets in Cancer"

_biology, 2021, doi:10.3390/biology10111178_

Round 1

Reviewer 1 Report

Authors Lim et al. explore the potential role of siglecs as an immune checkpoint inhibitor and possible therapeutic oncologic target.

While there has been recent interest in these motifs in various human diseases, thus far there are no publications exclusively summarizing their role in cancer

The authors do an exhaustive job in exploring the biological function of each siglec subtype and their current implication in cancer pathophysiology and therapeutics.

Once the minor points below are addressed I recommend this paper for publication.

Minor points:

  • I do not see the relevance of the first part of the sentence within line 120 as it stands within the paragraph: “However, CD22 has been found to be lowly expressed on several human lung cancer cells [45]…” Would it be possible to summarize the expression of CD22 in other solid tumors?
  • Line 121-122: Please expand on possible reasons why “CD19/CD22 CAR-T cells can still lead to hemophagocytic lymphohistiocytosis-like toxicities.”
  • Line 179-181: Need a reference for the statement “As Siglec-5 in human cells is strongly repressed, it still remains unknown what exactly triggers Siglec-5 expression in T cells in cancers”
  • Line 220-221: Need a reference for the statement “High levels of Siglec-7 in intra-tumoral macrophages were also associated with poor outcome in vaccinated and metastatic colorectal cancer patients.”
  • Line 320: Please expand on which types of cancer patients “which is associated with poor prognosis in cancer patients”
  • Lines 454-457: Need a reference for the statement “Several bsAbs (AMG330, AMV564 and AMG673) targeting CD33+ AML are being evaluated in preclinical and early clinical trials to determine their efficacy as immunotherapeutic agents”
  • Lines 466-468: Need a reference for the statement “Although preliminary results from this study indicate that AMG330 showed potent anti-leukemic activity, their rapid renal drug clearance required continuous intravenous infusion (CIV) for up to several weeks. “ Refences will also been needed for most statements in the subsequent paragraph lines 469-482.
  • Lines 593-595: Need a reference for the statement “Additionally, the antitumor activity of Epirubicin-loaded liposomes decorated with sialic acid-cholesterol conjugate was found superior to Pix-loaded liposomes modified with sialic acid–octadecylamine conjugate.”

Author Response

We have attached a revised manuscript based on inputs from both Reviewers 1 and 2.

Minor points:

  • I do not see the relevance of the first part of the sentence within line 120 as it stands within the paragraph: “However, CD22 has been found to be lowly expressed on several human lung cancer cells [45]…” Would it be possible to summarize the expression of CD22 in other solid tumors?

We wanted to emphasis that CD22 as a target in CD19neg tumors is not commonly found in non-B cell malignancies. We have elaborated this point with a new paragraph.

Line 113: “However, even CD22 expression can be low or even null in several human lung cancer cells, pancreatic, prostate, breast and liver cancers to be targetable (https://www.proteinatlas.org/ENSG00000012124-CD22/pathology) [46]. “

  • Line 121-122: Please expand on possible reasons why “CD19/CD22 CAR-T cells can still lead to hemophagocytic lymphohistiocytosis-like toxicities.”

We have provided the immune signatures associated with hemophagocytic lymphohistiocytosis-like toxicities with a new paragraph.

Line 115: “More importantly, CD19/CD22 CAR-T cell therapy can lead to hemophagocytic lymphohistiocytosis-like toxicities due to persistent elevation of inflammatory cytokines in concert with pronounced NK-cell lymphopenia [47,48].“

  • Line 179-181: Need a reference for the statement “As Siglec-5 in human cells is strongly repressed, it still remains unknown what exactly triggers Siglec-5 expression in T cells in cancers”

Line 165: A reference is added: As Siglec-5 in human cells is strongly repressed, it still remains unknown what exactly triggers Siglec-5 expression in T cells in cancers [13].

  • Line 220-221: Need a reference for the statement “High levels of Siglec-7 in intra-tumoral macrophages were also associated with poor outcome in vaccinated and metastatic colorectal cancer patients.”

Line 200: A reference is added: High levels of Siglec-7 in intratumoral macrophages were also associated with poor outcome in vaccinated and metastatic colorectal cancer patients [93].

  • Line 320: Please expand on which types of cancer patients “which is associated with poor prognosis in cancer patients”

Line 289: It now reads, “which is associated with poor prognosis in patients with gastric, colorectal or ovarian cancers [144,145]”

  • Lines 454-457: Need a reference for the statement “Several bsAbs (AMG330, AMV564 and AMG673) targeting CD33+ AML are being evaluated in preclinical and early clinical trials to determine their efficacy as immunotherapeutic agents”

Line 467: It now reads, “Several bsAbs (AMG330, AMV564 and AMG673) targeting CD33+ AML are being evaluated in preclinical and early clinical trials (NCT02520427, NCT03144245, NCT03224819) to determine their efficacy as immunotherapeutic agents.”

  • Lines 466-468: Need a reference for the statement “Although preliminary results from this study indicate that AMG330 showed potent anti-leukemic activity, their rapid renal drug clearance required continuous intravenous infusion (CIV) for up to several weeks. “ Refences will also been needed for most statements in the subsequent paragraph lines 469-482.

Line 476: A reference is added: Although preliminary results from this study indicate that AMG330 showed potent anti-leukemic activity, their rapid renal drug clearance required continuous intravenous infusion (CIV) for up to several weeks [183].

Line 479-489: References have been added to this subsequent paragraph.

  • Lines 593-595: Need a reference for the statement “Additionally, the antitumor activity of Epirubicin-loaded liposomes decorated with sialic acid-cholesterol conjugate was found superior to Pix-loaded liposomes modified with sialic acid–octadecylamine conjugate.”

Line 585: A reference is added: Additionally, the antitumor activity of Epirubicin-loaded liposomes decorated with sialic acid-cholesterol conjugate was found superior to Pix-loaded liposomes modified with sialic acid–octadecylamine conjugate [212]

Reviewer 2 Report

In the review “Siglecs as Therapeutic Targets in Cancer” by Lim et al, the authors review current therapeutic strategies targeting these factors in cancer. This is an interesting topic, however, the manuscript can be improved.

I found the main topic of this review confusing. Authors remark several times in the abstract, summary, and text that the presence of abundant sialic acids in cancer cells helps them to evade immune responses, however, most of the review is dedicated to strategies that target siglecs in cancer cells and are not oriented to overcome immune evasion. Indeed, authors review all the siglecs present in immune cells, while the therapeutic strategies that they mention are mostly dedicated to targeting siglecs in cancer cells and only a very few of them are really focused on fighting immune evasion or targeting the immune system.

The summary and the abstract need improvement. The style is poor compared with the rest of the manuscript. Several sentences do not make sense. For example: “Today, our understanding of tumor immune evasion is, at least in part, driven through their interactions with immunoinhibitory Siglec receptors” or “Today, our increased understanding of these Siglec receptors, which are mostly immunoinhibitory are exploited by cancer cells, which have evolved to widely overexpress various sialic acid structures”. Also, the summary and abstract are misleading since they seem to convey that the main strategies that are going to be reviewed are focused on fighting immune evasion.

The figures are also problematic. They are not referred to in the text. Figure 2 has small fonts that are not readable, and it is not well explained. Panels A and B could be separated in a single figure from panel C. There is a nice figure dedicated to targeting siglecs in cancer cells. Including a figure focusing on targeting siglecs to modulate immune cells might be useful.

Author Response

  • I found the main topic of this review confusing. Authors remark several times in the abstract, summary, and text that the presence of abundant sialic acids in cancer cells helps them to evade immune responses, however, most of the review is dedicated to strategies that target siglecs in cancer cells and are not oriented to overcome immune evasion. Indeed, authors review all the siglecs present in immune cells, while the therapeutic strategies that they mention are mostly dedicated to targeting siglecs in cancer cells and only a very few of them are really focused on fighting immune evasion or targeting the immune system.

Lim: As blocking the Siglec-sialic acid axis can have beneficial effects in cancer therapy, strategies to target either the Siglec receptors or sialic acid ligands would diredtly/ indirectly fight immune evasion and subsequently reduce immunosuppression.

  • The summary and the abstract need improvement. The style is poor compared with the rest of the manuscript. Several sentences do not make sense. For example: “Today, our understanding of tumor immune evasion is, at least in part, driven through their interactions with immunoinhibitory Siglec receptors” or “Today, our increased understanding of these Siglec receptors, which are mostly immunoinhibitory are exploited by cancer cells, which have evolved to widely overexpress various sialic acid structures”. Also, the summary and abstract are misleading since they seem to convey that the main strategies that are going to be reviewed are focused on fighting immune evasion.

Lim: We have corrected both the summary and abstract to emphasis that this review gives an update on all 15 human siglecs (including non-sialic acid binding Siglec XII) and current targeted strategies to disrupt the siglec-sialic axis to favor antitumor response. These strategies target the surface Siglec-sialic acid interactions to result in tumor cell apoptosis.

  • The figures are also problematic. They are not referred to in the text. Figure 2 has small fonts that are not readable, and it is not well explained. Panels A and B could be separated in a single figure from panel C. There is a nice figure dedicated to targeting siglecs in cancer cells. Including a figure focusing on targeting siglecs to modulate immune cells might be useful.

Lim: We have updated Figure 2 to make Figures 2 and 4, elaborated on the figure caption and increased the font size. In addition, we have included a new section 2.16 and figure 3 to illustrate the downstream ITIM/ITAM signaling leading to immune cell responses. All the figures are now referenced in the text.

Round 2

Reviewer 2 Report

The authors have addressed the main concerns and the review is now suitable for publication